



# Transition process of abrupt climate change based on global sea surface temperature over the past century

Pengcheng Yan[1], and Wei Hou[2], Guolin Feng[1,2]

[1]College of Atmospheric Sciences, Lanzhou University, Lanzhou, 730000, China.
[2]National Climate Center, China Meteorological Administration, Beijing, 100081, China.

*Correspondence to*: G. L. Feng(fenggl@cma.gov.cn)

**Abstract.** We propose a new concept of abrupt climate change transition and create a novel detection method to identify the transition process. With this method, how the climate system transits from one stable state to another could be verified clearly. By applying this method to the global sea surface temperature data over the past century, several climate change
processes are detected, including their starting state(moment), persist time, and ending state(moment) etc. According to the spatial distributions, the locations of climate changes mainly occurred in Indian ocean and western Pacific before the middle twentieth century, while the climate changes in 1970s located in equatorial middle-eastern Pacific, and the climate changes happened in the middle and southern Pacific since the end twentieth century. In addition, an quantitative relationship among the transition process parameters has been exposed in theory and practice, which the relationship between the rate and
stability parameters is linear, and the relationship between the rate and change amplitude parameters is quadratic.

## 1 Introduction

Including a variety of factors, climate system is considered to be a gigantic and complex system, each member of the system following a certain rule, and there being a certain interaction between members. According to the previous work(Goldblatt et al, 2006; Alexander et al 2012; Baker and Charlson 1990; Charney and DeVore, 1979; Zerkle et al 2012), climate system has
two or more stable states, and the system transiting form one state to another is called climate change(Thom 1972; Lorenz 1963, 1976; Rial 2004). Climate change has an important impact on social politics and economic environment, and it is closely related to human survival, production, and life. The climate change issue has aroused great concern of the international community(IPCC 2014). When climate changes, the system jumps between different states, and it experiences a period(Yan et al 2012, 2013). Most traditional theories and detection methods(Wei 1999; Feng et al 2011; He et al 2012) of
climate change were focusing on the changing of statistics before and after climate change and ignored the duration, such as Yamamoto(Yamamoto et al 1986), Mann-Kendall(Mann 1945; Kendall et al 1955, 1976), Moving T-test, Moving Cut data-Approximate Entropy(He et al 2009; Jin et al 2015). Therefore, it is urgent to study the transition process of climate change. By understanding the climate change events form the angle of transition period, we can dig more phenomenon about climate change. Thus a new concept about transition process of climate change is proposed, and the detection method is created(Yan



et al 2015). By referring Fu's work(Fu et al 1992), climate change has 4 types: change in mean, change in variance, change in trend, and change of seesaw. While all 4 kinds of climate changes could be transformed to be change in mean by mathematical method, therefore this method mainly analyze the climate change in mean. With this method, five climate change processes of the Pacific Decadal Oscillation index was identified(Yan et al. 2015) and the climate change process of

500hPa temperature field was analyzed too(Yan et al 2014).

When climate system transits from one stable state to another via a process, the persist time could be indentified no matter how long it lasts. Muldelsee(2000) developed a regression technique to identify such process with a ramp function. In present paper, we use a traditional model(logistic model) to regress a real time sequence. The parameters of the model obtained via regressing step describe the climate change process. The mode could represent different degrees of change. We

also realize that climate change has a relationship with the length of time sequence(Yan et al. 2015), thus a sub-sequence is extracted from the entire sequence for regression. A group of parameters could be obtained when the sub-sequence moves to a different position. With the percentile threshold method to this parameters, we determine the climate changes with given threshold(98%). Based on the concept of climate change process, we detect several climate changes of Global Sea Surface Temperature(GSST) in the past century.

**2. Method and Data**

Details of the proposed method have been thoroughly discussed (Yan *et al*. 2015); a brief description follows. A biological model was created to study its complex dynamics (May 1976). The model also describes an abrupt change in mean (Liu *et al*. 2004) and can be expressed as:

$$\dot{x} = \kappa x(x - \mu).\tag{1}$$

Equation (1) is a function describing the model (solving this model requires rewriting the equation in its difference form $x(t+1) = x(t) + \tau \kappa x(t)(x(t) - \mu)$). Obviously, such a model describes a system which transitions from one state to another, and its two states are $x = 0, x = \mu$. To make the model handle the more general case in which the system transitions from one state ($x = v$) to another ($x = \mu$), it can be modified as follows:

$$\dot{x} = \kappa(x - \mu)(v - x).\tag{2}$$

The logistic model and its modified form have been used to study abrupt change in many fields (Guttal and Jayaprakash 2008). The physical meanings of the parameters ($\mu, v, \kappa$) were thoroughly discussed (Yan *et al*., 2014, 2015). This paper introduces how to estimate the parameters based on a time series.

A piecewise function is created to describe a curve which is similar to the preceding one and is divided into three sections:



$$x = \begin{cases} v & Section\,1 \\ h \cdot t + \xi & Section\,2 \\ \mu & Section\,3 \end{cases}. \tag{3}$$

In Sections 1 and 3, the system stays in two states, $x = v, x = \mu$. The parameters can be expressed as:

$$v = \frac{1}{n_1} \sum_{i=1}^{n_1} x_i, \quad \mu = \frac{1}{n_3} \sum_{i=n_2+1}^{n_1+n_2+n_3} x_i, \tag{4}$$

where $n_1$ is the persistence time in Section 1 and $n_3$ is the persistence time in Section 3.

5    In Section 2, the system is in transition from state $v$ to state $\mu$. Assuming that the transition process is linear, the slope of the process is defined as the rate of change. Based on two points of the process, $A(t_a, x_a)$ and $B(t_b, x_b)$, the parameter $h$ can be expressed as:

$$h = \frac{x_b - x_a}{t_b - t_a}. \tag{5}$$

The parameters $\alpha, \beta$ are defined to describe the two points' locations:

$$\begin{cases} x_\alpha = \alpha(\mu - v) + v \\ x_\beta = \beta(\mu - v) + v \end{cases}. \tag{6}$$

And the solution(as folloews) of the model could be used to describe the points' locations too.

$$t = \frac{1}{\kappa(\mu - v)} \ln\left( \frac{x_0 - \mu}{x_0 - v} \cdot \frac{x - v}{x - \mu} \right) + t_0, \tag{7}$$

Based in Eqs. (5-7), the parameter $h$ can be expressed as:

$$h = \frac{\alpha(\mu - v) - \beta(\mu - v)}{\dfrac{1}{\kappa(\mu - v)}\left( \ln\left( \dfrac{\alpha(\mu - v)}{\alpha(\mu - v) - \mu + v} \cdot \dfrac{\beta(\mu - v) - \mu + v}{\beta(\mu - v)} \right) \right)}$$
$$= \kappa(\mu - v)^2 \frac{\alpha - \beta}{\ln\left( \dfrac{\alpha}{\beta} \cdot \dfrac{1 - \beta}{1 - \alpha} \right)} = \kappa \omega^2 \chi \tag{8}$$

15    Which the new location parameter is defined as $\chi = \dfrac{\alpha - \beta}{\ln\left( \dfrac{\alpha}{\beta} \cdot \dfrac{1 - \beta}{1 - \alpha} \right)}$, and a change amplitude parameter is defined as

$\omega = \mu - v$. Yan $et\ al$ (2015) discovered that $\chi$ varies only slightly when the values of $\alpha, \beta$ fall within a certain range,

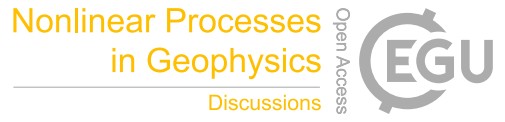

and when $\alpha = 0.2$, $\beta = 0.8$, $\chi \left( = 0.2164 \right)$ is constant.

According to Eq. (3), the parameter $h$ can be estimated by the least-squares method:

$$h = \sum_{i=n_1+1}^{n_1+n_2} \bar{t}_i \cdot \bar{x}_i \Big/ \sum_{i=n_1+1}^{n_1+n_2} \bar{t}_i^2 , \qquad (9)$$

where $n_2$ is the persistence time in section 2. Then parameter $\kappa$ can be expressed as:

$$\kappa = \frac{h}{\left( \mu - \nu \right)^2 \chi} . \qquad (10)$$

Based on Eqs. (4) and (9), parameters $\mu, \nu, h$ can be estimated optimally by changing $n_1, n_2, n_3$, and parameter $\kappa$ can be calculated using Eq. (10).

Note that parameter $\kappa$ is a stability parameter, which means that the larger its value, the more unstable the system becomes. The quantitative relationship among the rate of change $h$, the stability parameter $\kappa$, and the change amplitude $\omega$ is shown in Eq. (8). The relationship between the rate of change $h$ and the stability parameter $\kappa$ is linear, but that between the rate of change $h$ and the change amplitude $\omega$ is parabolic. According to a numerical test which applied this method to several ideal models, these relationships were clearly verified. The test showed that the ratio of the rate of change and the stability parameter is constant when the change amplitude is fixed. Moreover, when the stability parameter is fixed, the ratio of the rate of change and the square of the change amplitude is also constant.

Yan *et al*. (2015) applied this method to the Pacific decadal oscillation time series and verified that the transition process existed. In the present paper, this method is used to analyze the GSST transition process. Note that Yan *et al*. set the sub-sequence to test the Pacific decadal oscillation time series as 10a, 20a, 30a, and 40a, but here the sub-sequence is set to 20a to detect the abrupt change process(ACP) for GSST.

The dataset was reconstructed by the National Oceanic and Atmospheric Administration ( http://www.esrl.noaa.gov/psd/data/gridded/data.noaa.ersst.html). The time span of the monthly data is from January 1854 to November 2012, and the spatial resolution is 2×2°. These data have been determined to be reliable (Hirahara *et al*. 2014; Liu *et al*. 2015; Ratna 2015). During the calculation, the time series of each grid was processed for anomalies by month.

## 3 Spatial positions of abrupt changes

### 3.1 Abrupt changes at different start/end moments

The detection method is applied to identify the temperature sequence of each grid. In order to confirm the climate change, we count all the start moments and end moments. As shown in Fig.1, the probabilities of start moments and end moments are displayed from 1854 to 2010. According to start moment, the probability in 1878, 1942, 1976 and two periods(1890-1920, 1990-2010) is high (larger than 1%, as shown in Fig.1a), which indicates that most of the grid points started to change in this



periods. While according to end moment, the probability in 1886, 1950 and 1982 and two periods(1900-1930, 1990-2010) is high (larger than 1%, as shown in Fig.1b), which indicates that most of grid points end the change in this periods. By comparing the two time series, a certain corresponding relationship between the probabilities shows that the climate changes started in periods of Fig.1a and ended in periods of Fig.1b as marked by the red arrows and blue boxes respectively. As for

the climate change started in 1878 and ended in 1886, the frequency of former is larger than later. The reason maybe that some grids started to change in 1878, while some of this grids did not end in 1886. The same situation occurs in other periods.

In order to testify the corresponding relationship among the probabilities, the spatial distribution of abrupt change classified based on the start/end moment is shown in Fig.2.

(1) As shown in Fig. a, the abrupt change start in 1878 occurred mainly in the northern Indian Ocean, part of the central North Pacific and South Pacific, and the equatorial Atlantic region. These regions coincided with those of abrupt change ending in 1886 (Fig. b), which indicates that they belonged to the same ACP.

(2) As shown in Fig. c, the spatial distribution of abrupt change start in 1942 coincided with that end in 1950 (Fig. d), mainly covering the coastal regions in the northern and western Indian Ocean and part of the equatorial Atlantic regions. This

indicates that they are part of the same ACP.

(3) As shown in Fig. e, the abrupt change start in 1976 mainly occurred in the middle-eastern equatorial Pacific Ocean and small regions of the South Pacific near the South Pole. These regions completely coincided with those of the abrupt change ending in 1982 (Fig. f), indicating that they were part of the same ACP..

(4) As shown in Fig. g, the abrupt change start in 1890-1920 may be divided into three principle periods: that in 1890-1898

occurred mainly in the Indian and South Pacific Oceans; that in 1900-1993 occurred mainly in the North Pacific and Atlantic; and the most one in 1908-1909 occurred mainly in the eastern and western regions of the equatorial Pacific. This coincides with the spatial distributions (Fig. h) of abrupt change ending in 1902-1903, 1896-1898 and 1908-1910, and proves that they are part of the same ACP.

(5) As shown in Fig. i, the abrupt change which occurred in 1990-2010 may be divided into three periods: that in 1995-1997

occurred mainly in the western region of the South Pacific; that in 1998-1999 also occurred mainly in the western South Pacific and some regions of the North Pacific; and the most recent one occurred in 2005 and 2007, mainly in the Arctic region. In addition, the abrupt changes start in1995-1997 ended in 1997-1999; and that start in 1998-1999 and 2005-2007 in the Arctic region ended in 2006-2008 (Fig. j). These indicate that they were part of the same abrupt change.

The above analysis verifies that the abrupt change of a certain grid point undergoes three steps: start, sustaining and end. The

spatial distribution situation of the entire ACP coincides to a large degree with the distribution situation of the abrupt change "point" proposed by Xiao(2007). This indicates that abrupt changes detected by the traditional "abrupt change point" are contained within the "ACP".

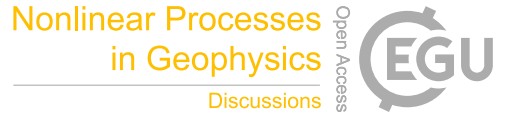

### 3.2 Time sequence variation of abrupt change

By studying the abrupt changes which occurred in 1878, 1942 and 1976 and two periods(1890-1920, 1990-2010), the averages of the spatial grid points of climate change are calculated. Seven sequences are obtained to describe the transition process as shown in Fig. 3.

In terms of sequence variation, the climate changes occurring before 1960 were characterized by low temperature. For that occurring in 1878, the temperature of the abrupt changes decreased by an average of 0.12℃; the temperatures of the three abrupt changes in 1890-1920 decreased by 0.11℃, 0.16℃ and 0.06℃,respectively; and that of the abrupt change in 1942 decreased by 0.08℃. The abrupt changes occurring after 1960 were characterized by temperature rise. The temperature of the abrupt change occurring in 1976 increased by 0.35℃, and that of the abrupt change in 1990-2010 increased by 0.11℃. In

addition, for the two abrupt changes occurring at the two periods (1890-1920 and 1990-2010), due to the fact that the distribution latitude was high, the average temperature of the sequence was about 8-10℃ lower than the average temperature of other three abrupt changes occurring at 1878, 1942 and 1976, which was about 20℃.

### 3.3 Spatial distribution of abrupt change duration

With the start and end moments of a certain abrupt change given in the method, the period of the abrupt change may be

determined. Based on this, the spatial distribution of the abrupt change duration is obtained as shown in Fig. 4 Deep azure signifies that the duration is no longer than 30 months; orange signifies that the duration is about 30-60 months; turquoise signifies that the duration is about 60-90 months; and magenta signifies that the duration is about 90-120 months.

According to the start moment, 8 figures of different period ACP are analyzed:

(1) For the abrupt change start in 1878 shown in Fig. a, the duration times of the abrupt changes occurring in areas of the

Northern Hemisphere were short, such in as the north Indian Ocean, central North Pacific and some regions of the North Atlantic, while those of the abrupt changes occurring in the Southern Hemisphere were long, such as in the central South Pacific.

(2) For the abrupt change start in 1896-1898 shown in Fig. b, mainly occurring in the south of the Southern Hemisphere, the duration times were all greater than 60 months. That in the Pacific Ocean was 60-90 months, and that in the southern Indian

Ocean was slightly longer.

(3) For the abrupt change start in 1900-1903, shown in Fig. c, mainly occurring in the Pacific Ocean, the overall duration time was long (more than 60 months); and that in the eastern part was 90-120 months, longer than that in the western part, which was 60-90 months.

(4) For abrupt change start in 1908-1909, shown in Fig. d, mainly occurring in parts of the coastal regions of the Pacific and

southern Indian Oceans, the regions were small, while the overall duration time was long, all being longer than 60 months.

(5) For the abrupt change start in 1942, shown in Fig. e, the overall duration times were long throughout the world, especially in the coastal regions of the western Indian Ocean and central Atlantic, both being about 90-120 months.



(6) For the abrupt change start in 1976 occurring mainly in the equatorial middle Pacific, shown in Fig. f, the duration time was 60-90 months.

(7) The duration time of abrupt change start in 1989-1999 was relatively short, as shown in Fig. g. The duration time in the polar regions and western Pacific Ocean were shorter than 60 months, while that in the middle-east equatorial Pacific was a little longer, being longer than 60 months.

(8) For the abrupt change start in 2005-2006, shown in Fig. h, mainly occurring in the Arctic region, the duration time was less than 30 months.

Through the above analysis, the following observations are made: The duration times of abrupt change before the 1960s were 60-120 months, and most were longer than 90 months; those in the 1970s were 60-90 months; those in the 1980s and 1990s were mostly 30-60 months; and those at the start of the 21st century were shorter than 30 months. Therefore, it is understood that the duration time of abrupt change under the background of global warming has shortened.

## 3.4 Spatial distribution of abrupt change amplitude parameter

The different degree of abrupt change amplitude in different period are shown in Fig. 5: deep azure signifies that the cooling amplitude is larger than 2℃; orange signifies that the cooling amplitude is less than 2℃; turquoise light red signifies that the warming amplitude is less than 2℃; and magenta signifies that the warming amplitude is greater than 2℃.

According to the start moment, 8 abrupt changes of different period are analyzed:

(1) For the abrupt change start in 1878, shown in Fig. a, only in the northern region of South Pacific did the temperature increase slightly; those in the other regions all decreased, and the amplitude was less than 2℃.

(2) The abrupt change start in 1890-1920 is shown in Figs. b, c and d; the temperature decreased in most areas, while in some it increased. The specific situations are as follows: For the abrupt change start in 1896-1897, shown in Fig. b, the temperature increased by a small degree only in the south of the South Pacific; it decreased in all other regions, and the amplitude was less than 2℃; for the abrupt change start in 1900-1993, shown in Fig. c, the temperature decreased overall, and the decrease amplitude in the eastern region of the North Pacific was large (larger than 5℃); for the abrupt change start in 1908-1909, shown in Fig. d, the temperatures decreased mainly in the equatorial regions, and increased in the eastern and western regions of the North Pacific and western region of the South Pacific.

(3) For the abrupt change start in 1942, shown in Fig. e, the temperatures in the coastal region of the western Indian Ocean mostly decreased (the amplitude was less than 2℃); and those in some regions of the North Pacific mostly increased.

(4) Fig. f is the abrupt change start in 1976; the temperatures throughout the entire region increased, and the amplitude was large (larger than 2℃).

(5) Figs. g and h show the abrupt changes start in 1990-2010. The temperatures increased almost everywhere throughout the world, and decreased only in some regions of the South Pole start in 1989-1999. The temperatures in other regions all increased, and the amplitude in the central regions of equatorial Pacific was larger than 2℃. The temperature of abrupt

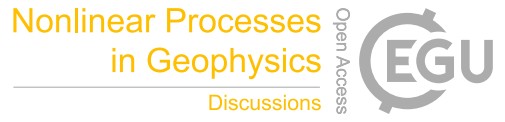

change start in 2005-2006 mainly increased in the Arctic region.

Based on the above analysis, it is believed that the temperatures of abrupt changes before the 1960s mostly decreased; after experiencing the temperature rise in the 1970s, the temperatures of abrupt change throughout the world mostly increased.

## 4 Statistical Characteristics of Parameters

### 4.1 Multistability/bistability state characteristics of abrupt climate change system

The difference between different states of the ACP is counted in Fig. 6, and Fig. a shows the statistics of the start states. It is observed that the statistical probabilities of 2 states are high, which indicates that these states are stable. The statistics of the end states are shown in Fig. b, and the statistical probabilities of the multiple states are large, which indicates that multiple stability states appear after abrupt changes of the system. After further study of the abrupt change amplitude parameter of the system, as shown in Fig. c, it is observed that the change amplitude parameter of the system presents a double-peak structure, and the two peaks are distributed symmetrically around the "zero point". This indicates that the system shifts in different states, and abrupt change is formed. Scholars have previously achieved similar conclusions in the study of climate system theories, i.e. the climate system is a multistability system(Baker. 1990; Alexander. 2012). The results shown in Fig. 7 further prove that bistability or multistability states exist in the sea surface temperature.

### 4.2 Quantitative relationship among abrupt change rate, abrupt change amplitude and instability parameter

The quantitative relationship among abrupt change rate $h$, instability parameter $\kappa$ and abrupt change amplitude parameter $\omega$ has previously been theoretically analyzed, and testified by the theoretical model. In this paper, by taking advantage of parameters based on the global SST sequence over the past 100 years, this conclusion is further proven using actual data. In view of the abrupt changes in 1878, 1942, 1976, in 1890-1920 and 1990-2010, the distribution relationships of parameters $h - \omega$ and $h - \kappa$ of each period sequence are respectively shown in Figs. 7 and 8 successively.

Fig. 7 shows the relationship between abrupt change rate $h$ and instability $\kappa$. Figs. a-e are the abrupt changes in different periods. Taking the abrupt change of 1942 in Fig. b as an example, the horizontal axis is the instability $\kappa$, and the vertical axis is the abrupt change rate $h$. With the increase of instability parameter, the abrupt change rate increases with a direct proportion, showing a direct proportion between the two. In terms of the numerical value of successive abrupt change, the instabilities of abrupt change in 1878, 1942 and 1976 are within the range from -0.15 to +0.15. The three abrupt changes all occurred in regions of middle and low latitude. However, for the abrupt changes in 1890-1920 and 1990-2010 occurring in regions of high latitude, the instabilities are within the range from -4 to +4. This indicates that the system stability of high latitude areas is lower than that of low latitude areas.

Fig. 8 shows the relationship between abrupt change rate $h$ and abrupt change amplitude parameter $\omega$. Taking the abrupt changes in 1990-2010 in Fig. e as an example, the horizontal axis is the abrupt change amplitude parameter, and the vertical

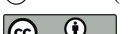



axis is the abrupt change rate. The rate increases with the increase of change amplitude parameter, and presents a parabolic increase. This indicates that the increase of abrupt change rate is doubled with the increase of change amplitude parameter. This also explains a phenomenon that the faster the speed of the abrupt change is, the larger the change amplitude parameter is, causing immense destructive effects on the environment.

The results of above the Figs. 7-8 prove the quantitative relationship provided in Formula (8), i.e. the abrupt change rate and instability are in direct proportion, and the abrupt change rate is in direct proportion with the quadratic abrupt change amplitude parameter. Based on this relationship, the existing abrupt change amplitude parameter of the actual climate sequence can be used to estimate the abrupt change rate and further estimate the stability of the system. It is also noticed that instability is a parameter related to the property of the system itself, and with a large amount of observed statistical data the
range may be obtained. This may then be used in the estimation of abrupt change of the system, and for the further estimation of future abrupt change amplitude parameter.

## 5 Conclusions and Expectations

The climate system is a gigantic, complex, and chaotic system with bistable and multistable states. The climate system transiting from one state to another is considered to be an abrupt climate change. The phenomenon of climate change has
been verified many times over the past 100 years, but studies focusing on abrupt change processes are rare. Therefore, based on GSST time-series data, a novel detection method has been used to study the abrupt change process in real time series. The results provide an understanding of the transition period, in which the system has already left its original state, but has still not reached its new state.

(1) The transition process is a significant period of an abrupt change event. According to the starting and ending times, an
abrupt change event can be divided into three periods. Several abrupt changes have been identified in the past 100 years based on GSST time series. Then the spatial distribution of each abrupt change was illustrated, revealing that abrupt change amplitudes in low-latitude regions are small, but those in regions of middle and high latitude are large.

(2) In view of these characteristics of the abrupt change process, the spatial distribution of abrupt change duration was analyzed. It was discovered that abrupt changes in different periods had characteristic durations. The durations of abrupt
changes before 1942 were generally long (about 5–10 years); that in 1976 was about 5–8 years; and those after this time were short (about 5 years). This indicates that against the background of global warming, the durations of abrupt change events have been shortened.

(3) The spatial distribution of abrupt change amplitudes indicates that before 1942, temperature mostly decreased throughout the world during these events, but mostly increased after that. The statistical results for abrupt change amplitude proved that
the SST is a bistable system and that the system state variables have shifted many times from one stable state to another.

(4) By analyzing the parameters estimated from the SST system, the quantitative relationships among system rate of abrupt change, change amplitude, and stability were studied and demonstrated that the relationship between the rate of abrupt



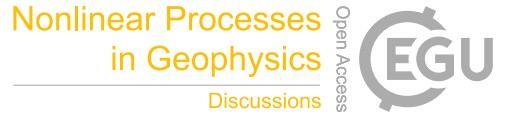

change and stability is linear, but the relationship between the rate of abrupt change and change amplitude is quadratic. This observation reveals the cause of the high rate and large amplitude of abrupt climate change.

In present paper, we have studied the transition process of abrupt climate change, based on reanalysis data from NOAA for the period 1854-2012, and the quantitative results show a bright prospect. With the quantitative relationship among the parameters of the transition process, a deeper research about the prediction of the abrupt climate change needs to be further studied.

**Acknowledgements.** This study was jointly sponsored by the National Natural Science Foundation of China (Grants 41305056, 41530531, 41375069) and the China Special Fund for Meteorological Research in Public Interest (Major projects) (Grant GYHY201506001).

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

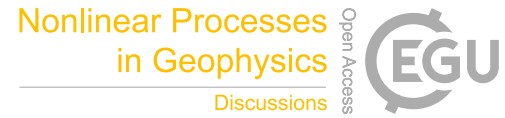



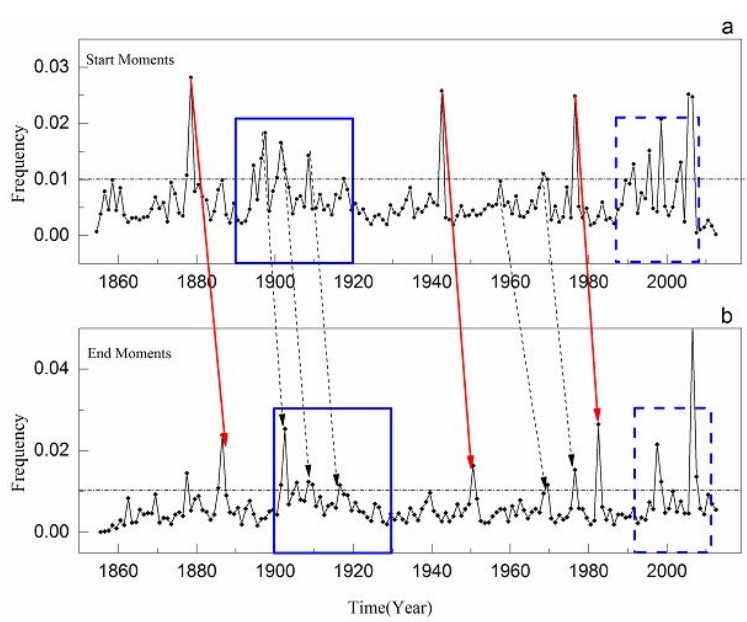

**Figure 1: Frequency of climate changes based on start moments(a) and end moments(b)**

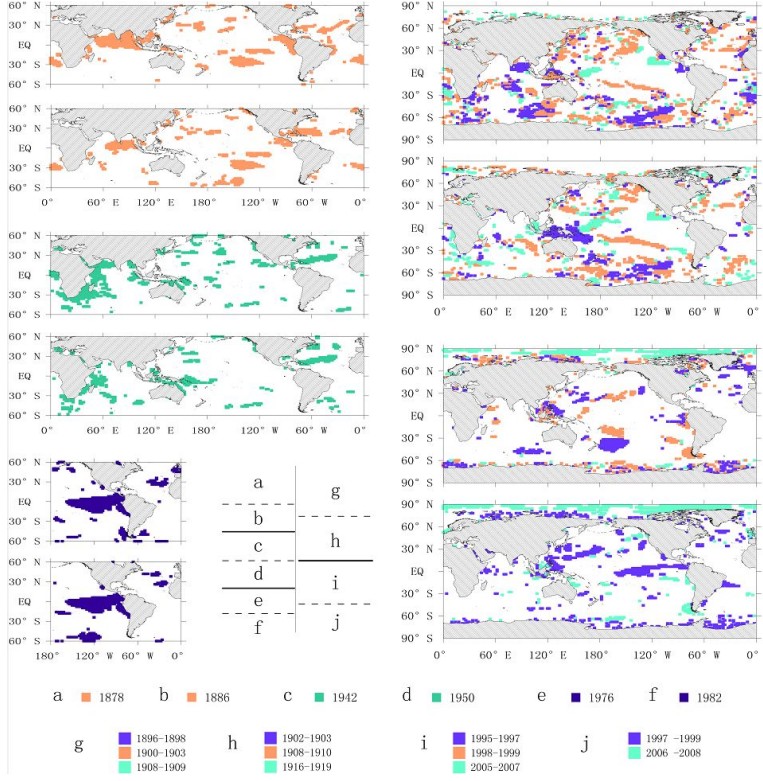



**Figure 2: Spatial distribution classified based on probability of start and end moments of abrupt change. Figures. a and b are the climate changes start in 1878 and ending in 1886. Figures. c and d are the climate changes start in 1942 and ending in 1950. Figures. e and f are the climate changes start in 1976 and ending in 1982. Figures. g and h are the climate changes start in 1890-1920 and ending in 1990-2010. Figures. i and j are the climate changes start in 1990-2010 and ending in 1990-2010.**

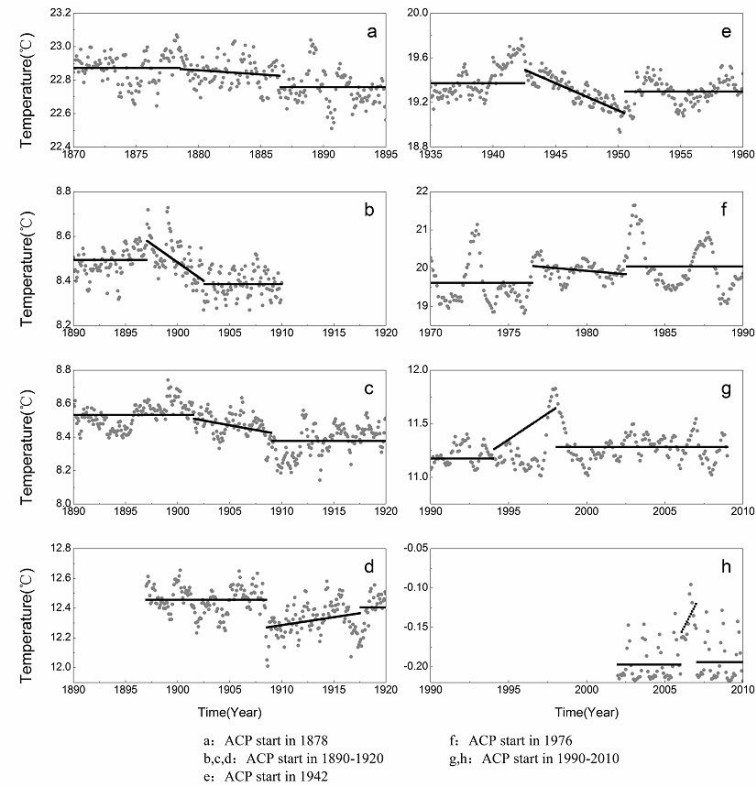

**Figure 3: Time sequence average of abrupt changes occurring under the background of the same abrupt change: Figure. a is the abrupt change start in 1878; Figures. b-d is the abrupt change occurring in 1890-1920, including the following; Figure. e is the abrupt change start in 1942; Figure. f is the abrupt change start in 1976; and Figures. g-h is the abrupt change start in 1990-2010.**





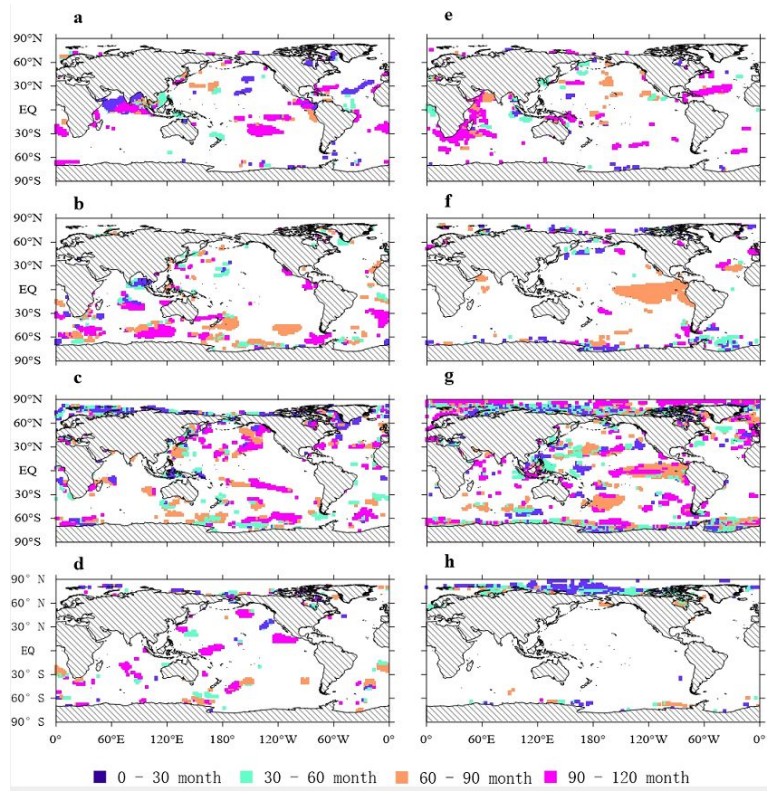

**Figure 4: Spatial distribution of abrupt change duration in different periods: Figure. a is the abrupt change in 1878; Figure. b is the abrupt change in 1896-1898; Figure. c is the abrupt change in 1900-1903; Figure. d is the abrupt change in in 1908-1909; Figure. e is the abrupt change in 1942; Figure. f is the abrupt change in 1976; Figure. g is the abrupt change in 1989-1999; and Figure. h is the abrupt change in 2005-2006.**

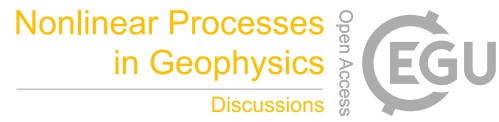



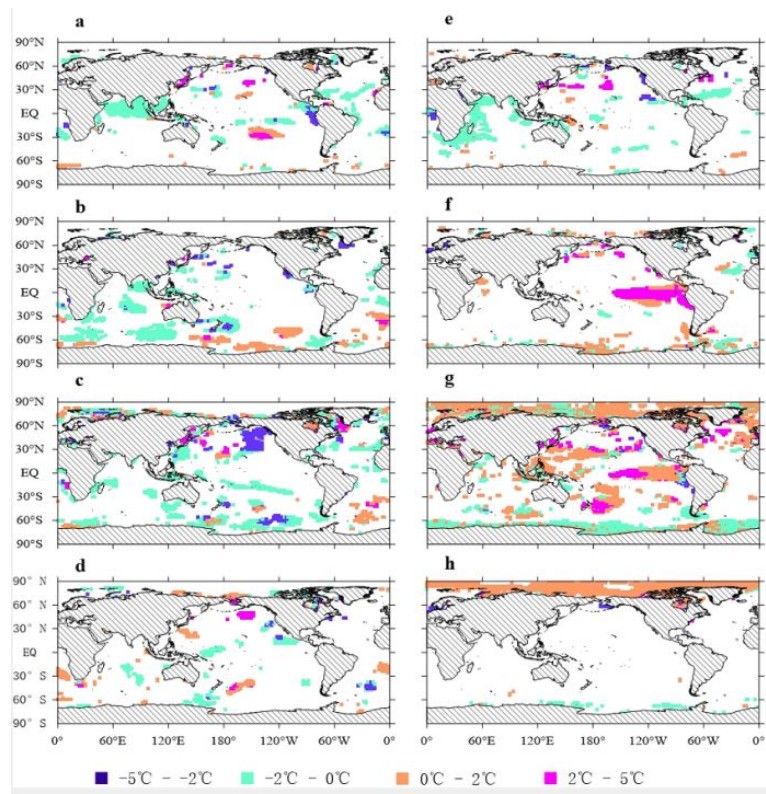

**Figure 5: Amplitude distributions of abrupt changes in different periods: Figure. a is the abrupt change in 1878; Figure. b is the abrupt change in 1896-1898; Figure. c is the abrupt change in 1900-1903; Figure. d is the abrupt change in in 1908-1909; Figure. e is the abrupt change in 1942; Figure. f is the abrupt change in 1976; Figure. g is the abrupt change in 1989-1999; and Figure. h is the abrupt change in 2005-2006.**

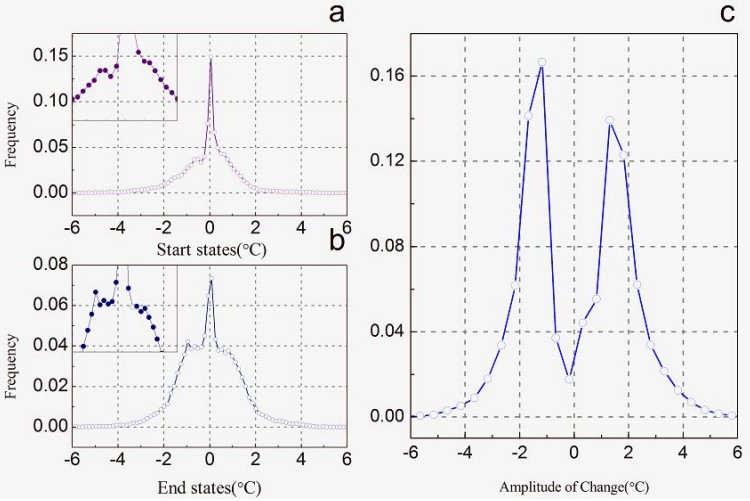

**Figure 6: Bistability state characteristics study of SST system: Figures. a and b are the system state variables before and after abrupt change, respectively; and Figure. c is the variation quantity in the ACP of the system, namely abrupt change amplitude**

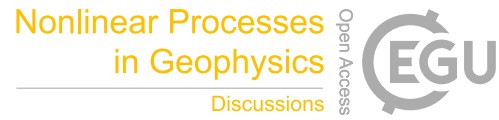



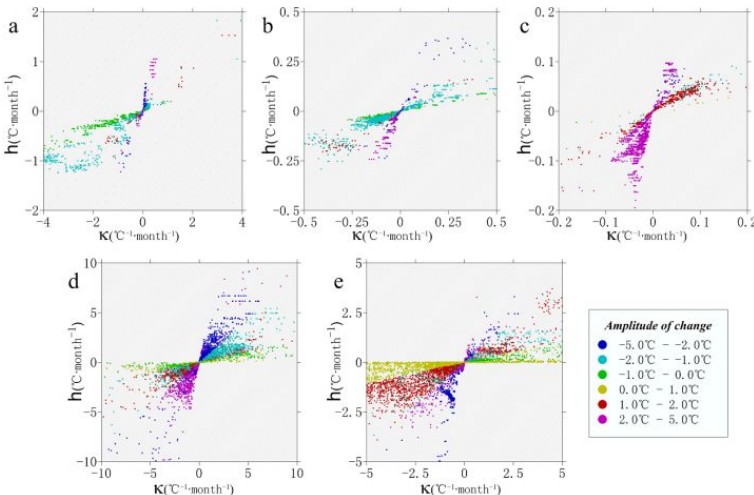

**Figure 7: Relationships between abrupt change rate $h$ and instability parameter $\kappa$ in different periods and different color represent different degree by abrupt change amplitude parameter. Figures. a-c show the abrupt changes in 1878, 1942 and 1976, respectively; and Figures. d-e show the abrupt changes in 1890-1920 and 1990-2010.**

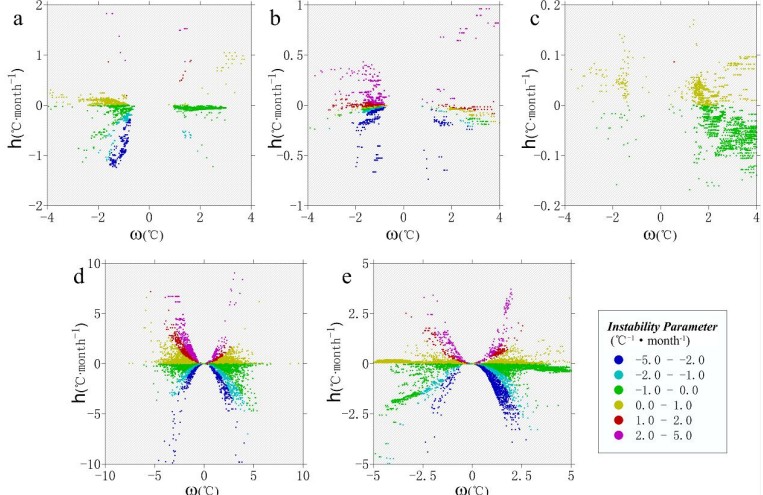

**Figure 8: Relationship between abrupt change rate $h$ and abrupt change amplitude parameter $\omega$, and different color represent different degree by instability amplitude. Figures. a-c show the abrupt changes in 1878, 1942 and 1976. Figures. d-e are the abrupt changes in 1890-1920, 1990-2010.**