# Peer review of "Transition process of abrupt climate change based on global sea surface temperature over the past century"

_Nonlinear Processes in Geophysics, 2016_

## Referee Comment (RC1) · Anonymous Referee #1 · 21 Feb 2016

This study reports an investigation on the abrupt change in global SST over the past hundred years using the method developed by the authors of this manuscript. There are a number of similar studies in abrupt changes of global SST, but they almost did not refer the transition processes involved. Several abrupt changes have been detected in the SST, and meanwhile the properties of the changes are discussed, by which some of the changes are identified as the cusp catastrophe with a gradually decreasing duration in the process. The statistics shows that the potential function of global SST is likely a bi-stable system. The most import findings in this study is that there is approximately linear relationship between the rate and stability parameters, in comparison with the quadratic one between the rate and the magnitude of the change. It is helpful in understanding the characteristics of the abrupt changes in climate science. I recommend considering the publication of this manuscript after making a further improvement in the English writing and correct some of mistakes in it. The detail points are as following terms:

(1) In figure 4, the abrupt changes occurred in different locations of the oceans at the same time. Please discuss possible mechanism about them. Are there some teleconnections between them or just a casual one?

(2) Since the parameter $\chi$ is a function of $\alpha$ and $\beta$, X becomes a constant if set$\alpha$=0.2, $\beta$=0.8. Why?

(3)The word "and" should be removed in the author list.

(4)Formula (3): The word "section1, section2, section3" may be replaced by "stage 1, stage2, stage3" or "domain1, domain2, doman3"?

Please also note the supplement to this comment:
http://www.nonlin-processes-geophys-discuss.net/npg-2016-7/npg-2016-7-RC1-supplement.pdf

---

## Author Comment (AC1) · 24 Feb 2016

Dear Referee,

We appreciate your interest to our article and your comments. We reply on the comments as follows:

(1)Referee. In figure 4, the abrupt changes occurred in different locations of the oceans at the same time. Please discuss possible mechanism about them. Are there some teleconnections between them or just a casual one?

Authors. Actually, it is a question about the standard of the abrupt change. In present manuscript, the method is based on the percentile threshold method, and this is a

relatively standard. Thus, when an abrupt change in one grid is detected, it means that the difference between the two states before and after the abrupt change of this moment has the biggest amplitude by comparing with the other moment. Taking fig. A3 for example, we know that both the two series have two abrupt changes in moment t1 and t2 respectively. However, when we detect the abrupt change by setting a standard, only one abrupt change can be considered as a real abrupt change. The amplitude of the abrupt change should be bigger than any others. Then, it is in moment t1 for grid 1 and it is in moment t2 for grid 2. Therefore, when the abrupt changes of different grids are detected, the abrupt change moments of two adjacent points could be different. It is inferred that the longer the time sequence, the more disperse the abrupt change space will be. There could be some teleconnections between the abrupt changes occurring in different locations at the same time, because they are likely to be driven by the same force. It's complex and difficult to explain the mechanism. Before that, it has to dig out how the different standard effect on the detection of abrupt change. When all the real abrupt changes are obtained, it is easy to know how the abrupt change transforming from one place to another in dynamics. However, in present manuscript, we just did some basic research about the transition process, proposing a special quantitative relationship. The relationship will be used to study the physical mechanism of abrupt change by climate model.

(2) Referee. Since the parameter [x] is a function of [a] and [b], [x] becomes a constant if set [a]=0.2, [b]=0.8. Why?

Authors. This issue was discussed in the previous paper(Yan et al, 2015). [a] and [b] represent two points(A(X[a],t[a]), B(X[a],t[b])) of the transition process, respectively (as shown in fig. A1), then , 0<[a],[b]<1. According to the function as mentioned in line 15, page 3, the location parameter [x] can be expressed by [a] and [b]. As shown in fig. A2, it is found that parameter [x] is almost constant when the values of [a] and [b] are within a certain range. Therefore, we think that that parameter [x] is approximately constant.

(3)Referee. The word "and" should be removed in the author list.

Authors. We are going to correct this mistake.

(4)Referee. Formula (3): The word "section1, section2, section3" may be replaced by "stage 1, stage2, stage3" or "domain1, domain2, doman3"?

Authors. The words "section1, section2, section3" will be replaced by "stage 1, stage2, stage3".

Appendix:

Figures

Fig. A1. The transition process of abrupt change with two points. (according to reference [1])

Fig. A2. The relationship parameter and parameters , , where the x axis is parameter , the y axis is parameter and the contour is parameter . (according to reference [1])

Fig. A3. To detect the abrupt change of different grids.

Reference [1] Yan P. C., Feng G. .L, Hou W., 2015: A novel method for analyzing the process of abrupt climate change. Nonlin. Processes Geophys. 22, 249–258, doi,10.5194/npg-22-249-2015

Please also note the supplement to this comment:
http://www.nonlin-processes-geophys-discuss.net/npg-2016-7/npg-2016-7-AC1-supplement.pdf
* * *
[Figure]

**Fig. 1.**

[Figure]

**Fig. 2.**

[Figure]

**Fig. 3.**

**Supplement:**

Dear Referee,

We appreciate your interest to our article and your comments. We reply on the comments as follows:

(1) **Referee.** In figure 4, the abrupt changes occurred in different locations of the oceans at the same time. Please discuss possible mechanism about them. Are there some teleconnections between them or just a casual one?

  **Authors.** Actually, it is a question about the standard of the abrupt change. In present manuscript, the method is based on the percentile threshold method, and this is a relatively standard. Thus, when an abrupt change in one grid is detected, it means that the difference between the two states before and after the abrupt change of this moment has the biggest amplitude by comparing with the other moment. Taking fig. A3 for example, we know that both the two series have two abrupt changes in moment t1 and t2 respectively. However, when we detect the abrupt change by setting a standard, only one abrupt change can be considered as a real abrupt change. The amplitude of the abrupt change should be bigger than any others. Then, it is in moment t1 for grid 1 and it is in moment t2 for grid 2. Therefore, when the abrupt changes of different grids are detected, the abrupt change moments of two adjacent points could be different. It is inferred that the longer the time sequence, the more disperse the abrupt change space will be.

  There could be some teleconnections between the abrupt changes occurring in different locations at the same time, because they are likely to be driven by the same force. It's complex and difficult to explain the mechanism. Before that, it has to dig out how the different standard effect on the detection of abrupt change. When all the real abrupt changes are obtained, it is easy to know how the abrupt change transforming from one place to another in dynamics. However, in present manuscript, we just did some basic research about the transition process, proposing a special quantitative relationship. The relationship will be used to study the physical mechanism of abrupt change by climate model.

(2) **Referee.** Since the parameter χ is a function of α and β, χ becomes a constant if set α=0.2, β=0.8. Why?

  **Authors.** This issue was discussed in the previous paper(Yan et al, 2015). α and β represent two points($A(X_\alpha, t_\alpha)$, $B(X_\beta, t_\beta)$) of the transition process, respectively (as shown in fig. A1), then α, β ∈ (0,1). According to the function as mentioned in line 15, page 3, the location parameter χ can be expressed by α and β. As shown in fig. A2, it is found that parameter χ is almost constant when the values of α and β are within a certain range. Therefore, we think that that parameter χ is approximately constant.

(3)**Referee.** The word "and" should be removed in the author list.

  **Authors.** We are going to correct this mistake.

(4)**Referee.** Formula (3): The word "section1, section2, section3" may be replaced by "stage 1, stage2, stage3" or "domain1, domain2, doman3"?

  **Authors.** The words "section1, section2, section3" will be replaced by "stage 1, stage2, stage3".

**Appendix:**

**Figures**

[Figure]

Fig. A1. The transition process of abrupt change with two points. (according to reference [1])

[Figure]

Fig. A2. The relationship parameter $\chi$ and parameters $\alpha$, $\beta$, where the x axis is parameter $\alpha$, the y axis is parameter $\beta$ and the contour is parameter $\chi$. (according to reference [1])

[Figure]

Fig. A3. To detect the abrupt change of different grids.

**Reference**

[1] Yan P. C., Feng G. .L, Hou W., 2015: A novel method for analyzing the process of abrupt climate change. Nonlin. Processes Geophys. 22, 249–258, doi,10.5194/npg-22-249-2015

---

## Referee Comment (RC2) · Anonymous Referee #2 · 25 Feb 2016

This work apply a previously published method to detect abrupt climate changes using a logistic model. As the authors state this kind of models are profusely used to study dynamical systems that have two different stable states and are able to change and jump from one state to the other. In a previous work in this same journal the authors have applied this method to study variability of Pacific Decadal Oscillation. As the name of this mode of variability of SST in Pacific suggest this oscillation is a system that are continually changing its behaviour between positive and negative states and therefore can be studied with the method proposed by the authors and conclusions are interesting in terms of climate variability and change. But to my viewpoint we can not apply the same kind of reasoning examining every single point in the ocean.

[Figure]

I suggest some additional work previous to the publication of the paper:

1 . PDO index synthesize the collective behaviour of an extended area of the ocean. But having a great number of series varying we can randomly detect a possible abrupt climate change that is not real in a single point of the ocean. To avoid this possibility authors put a threshold of 1% of the points performing this abrupt changes. Nevertheless the value of SST of one grid square is not independent of the value of the SST of neighbouring grid squares. Therefore I suggest to made an interdependence test to calculate this threshold. This can be done by replacing SST series with a Gaussian noise series generated from a normal population whose mean and variance are identical to that of the series over the whole studied period, examining abrupt changes in these series and repeating the process a number of times equal to the number of grid points. In this way the number of abrupt changes can diminish and the results could be more representative of real abrupt changes.

2 . Once performed the point 1 it is possible that the number of abrupt changes diminishes. In any case I suggest to the authors to made the effort to give an explanation of the patterns in terms of climate systems. Thus for example in figure 1 the changes detected in 1976 and 1982 are related to ENSO area. The change of 1976 is a well documented phenomenon called climate shift. In 1982 one of the most important ENSO episodes took place.

3 . As a minor question I also suggest to carefully read the paper because there are some important mistakes.

---

## Author Comment (AC2) · 11 Mar 2016

Dear Referee,

We appreciate your interest to our article. As your noticed, the method was published in this same journal, and this manuscript is a following work. In the previous paper, a novel method was proposed and applied to the Pacific Decadal Oscillation index. It's true that the index states "a system that are continually changing its behaviour between positive and negative states". By applying the detection method, several abrupt changes were verified. In the reference(Yan et al, 2014), this method was used to study the transition process of 500 hPa temperature field. That result exposed a new statistical characteristics of the abrupt climate change.

[Figure]

And for the comments, we reply on them as follows:

1.Referee. PDO index synthesize the collective behaviour of an extended area of the ocean.But having a great number of series varying we can randomly detect a possible abrupt climate change that is not real in a single point of the ocean. To avoid this possibility authors put a threshold of 1% of the points performing this abrupt changes. Nevertheless the value of SST of one grid square is not independent of the value of the SST of neighbouring grid squares. Therefore I suggest to made an interdependence test to calculate this threshold. This can be done by replacing SST series with a Gaussian noise series generated from a normal population whose mean and variance are identical to that of the series over the whole studied period, examining abrupt changes in these series and repeating the process a number of times equal to the number of grid points. In this way the number of abrupt changes can diminish and the results could be more representative of real abrupt changes.

Authors. As shown in figure 1(appendix), a numerical experiment is used to test the threshold . According to referee's suggestion, each original SST series is replaced by a new time series which is rebuilt by shuffle algorithm. The new time series is random, and its mean/variance is identical to the original one. Then, by detecting the abrupt changes of new series, the frequencies of start moment and end moment like figure 1 of manuscript are displayed. There is no moment at which the frequency is more than 1%. Another 10 experiments(the calculation cost 10 hours for each experiment) are taken as shown in figure 2, and it remains that nearly no moments at which the frequency is over 1%. Therefore, the threshold was set as 1% for SST series. Some explanation about the threshold is added in the manuscript.

2.Referee. Once performed the point 1 it is possible that the number of abrupt changes diminishes. In any case I suggest to the authors to make the effort to give an explanation of the patterns in terms of climate systems. Thus for example in figure 1 the changes detected in 1976 and 1982 are related to ENSO area. The change of 1976 is a well documented phenomenon called climate shift. In 1982 one of the most important

ENSO episodes took place.

Authors. The change of 1976 is known as a climate shift in the previous work, and it's an abrupt change "point". The abrupt changes of other periods also have similar background. Some explanations in terms of climate systems are added in section 3.1 of the manuscript as follows:

"It's obvious that the abrupt changes start in 1878 and 1942 mainly occurred in Indian ocean. And the Indian Ocean Dipole(IOD) of this two periods got strong negative phase and strong positive phase respectively(Suryachandra et al, 2002), which could be the trigger of the abrupt changes. The abrupt change start in 1976 is known as a climate shift, and most climate elements were detected to experience abrupt changes. The abrupt changes of 1890-1920 occurred mainly in Pacific ocean, and it is associated with Interdecadal Pacific Oscillation(IPO) index which transits from negative phase to positive phase during this period. The abrupt changes start in end of 2000s mainly occurred in high latitudes, which leads to a significant increase of temperature in the polar."

3.Referee. As a minor question I also suggest to carefully read the paper because there are some important mistakes.

Authors. The authors thank the anonymous referee, and some mistakes are corrected.

———————————————

Appendix:

Figure 1. Frequencies of the start moment and end moment based on an ideal numerical experiment.

Figure 2. Frequencies as figure 1, but 10 ideal experiments.

Correction about the manuscript

A new edition of the manuscript is submitted as a supplement, some modification about

the new edition:

1. Some mistakes mentioned by the referees are corrected;

2. Figure 1 is replaced by a new edition;

3. One paragraph about the climate background mentioned by referee #2 is added in section 3.1;

4. Two references are added in the manuscript;

5. Some other mistakes are corrected in the manuscript.

Please also note the supplement to this comment:
http://www.nonlin-processes-geophys-discuss.net/npg-2016-7/npg-2016-7-AC2-supplement.pdf

———————————————————————

[Figure]

**Fig. 1.**

[Figure]

**Fig. 2.**

**Supplement:**

**Transition process of abrupt climate change based on global sea surface temperature over the past century**

Pengcheng Yan[1], Wei Hou[2], Guolin Feng[1,2]

[1]College of Atmospheric Sciences, Lanzhou University, Lanzhou, 730000, China.

5 [2]National Climate Center, China Meteorological Administration, Beijing, 100081, China.

*Correspondence to*: G. L. Feng(fenggl@cma.gov.cn)

**Abstract.** A new detection method has been proposed to study the transition process of abrupt climate change. With this method, the climate system transiting from one stable state to another can be verified clearly. By applying this method to the global sea surface temperature over the past century, several climate changes and their processes are detected, including the

10 start state(moment), persist time, and end state(moment) etc. According to the spatial distributions, the locations of climate changes mainly occurred in Indian ocean and western Pacific before the middle twentieth century, while the climate changes in 1970s located in equatorial middle-eastern Pacific, and the climate changes happened in the middle and southern Pacific since the end twentieth century. In addition, an quantitative relationship of the transition process parameters is verified in theory and practice, (1) the relationship between the rate and stability parameters is linear, (2) the relationship between the

15 rate and change amplitude parameters is quadratic.

**1 Introduction**

Including a variety of factors, climate system is  a gigantic and complex system. Each member of the system follows a certain rule, and there is a certain interaction between members. According to the previous work(Goldblatt et al, 2006; Alexander et al 2012; Baker and Charlson 1990; Charney and DeVore, 1979; Zerkle et al 2012), climate system has two or

20 more stable states, and the system transiting form one state to another is called climate change(Thom 1972; Lorenz 1963, 1976; Rial 2004). Climate change has an important impact on social politics and economic environment, and it is closely related to human survival, production, and life. Climate change issue has aroused great concern of the international community(IPCC 2014). When climate changes, the system jumps from one state to another, and it experiences a period(Li et al 1996, Yan et al 2012, 2013). Most traditional theories and detection methods(Wei 1999; Feng et al 2011; He et al 2012)

25 were focusing on the changing of statistics before and after climate change, such as Yamamoto(Yamamoto et al 1986), Mann-Kendall(Mann 1945; Kendall et al 1955, 1976), Moving T-test, Moving Cut data-Approximate Entropy(He et al 2009; Jin et al 2015), and the duration was ignored. Therefore, it is urgent to study the transition process of climate change to understand how the system changes abruptly. By understanding the climate change events form the angle of transition period, more phenomenon about climate change would be exposed. Thus a new concept about transition process of climate change

was proposed, and the detection method was created(Yan et al 2015). By referring Fu's work(Fu et al 1992), climate change has 4 types: change in mean, change in variance, change in trend, and change of seesaw. While all 4 kinds of climate changes could be transformed to be change in mean by mathematical method, therefore the climate change in mean are studied mainly by this method. Five climate change processes of the Pacific Decadal Oscillation index was identified(Yan et al. 2015) and the climate change process of 500hPa temperature field was analyzed too(Yan et al 2014).

When climate system transits from one stable state to another via a process, the persist time could be indentified no matter how long the period lasts. Muldelsee(2000) developed a regression technique to identify such process with a ramp function. In present paper, a traditional model(logistic model) is used to regress a real time sequence. The parameters of the model obtained by regressing the time series. The mode could represent different degrees of change. It's noticed that climate change has a relationship with the length of time sequence(Yan et al. 2015), thus a sub-sequence is extracted from the entire sequence for regression. A group of parameters can be obtained when the sub-sequence moves on the entire time sequence. By using the percentile threshold method to the parameters with given threshold(98%), the climate changes are determined. Based on the concept of climate change process, several climate changes of Global Sea Surface Temperature(GSST) are detected in the past century.

**2. Method and Data**

Details of the proposed method have been thoroughly discussed (Yan *et al*. 2015); a brief description is as follows. A biological model was created, and it's a complex system(May 1976). The model also describes an abrupt change in mean (Liu *et al*. 2004) and it's expressed as:

$$\dot{x} = \kappa x(x - \mu). \tag{1}$$

By rewriting the equation in its difference form $x(t+1) = x(t) + \tau \kappa x(t)(x(t) - \mu)$), this equation can be solved. According to the solution, this model describes a system which transitions from one state to another, and the two states are $x = 0, x = \mu$ respectively. To make the model handle the more general case in which the system transitions from one state ( $x = \nu$ ) to another ( $x = \mu$ ), it can be modified as follows:

$$\dot{x} = \kappa(x - \mu)(\nu - x). \tag{2}$$

The logistic model and its modified form have been used to study abrupt change in many fields (Guttal and Jayaprakash 2008). The physical meanings of the parameters ( $\mu, \nu, \kappa$ ) were thoroughly discussed (Yan *et al*., 2014, 2015). This paper introduces how to estimate the parameters based on a time series.

A piecewise function is created to describe a curve which is similar to the preceding one and is divided into three stages:

$$x = \begin{cases} v & \text{stage } 1 \\ h \cdot t + \xi & \text{stage } 2 \\ \mu & \text{stage } 3 \end{cases}. \tag{3}$$

In stages 1 and 3, the system stays in two states, $x = v, x = \mu$. The parameters can be expressed as:

$$v = \frac{1}{n_1} \sum_{i=1}^{n_1} x_i, \quad \mu = \frac{1}{n_3} \sum_{i=n_2+1}^{n_1+n_2+n_3} x_i, \tag{4}$$

where $n_1$ is the persistence time in stage 1 and $n_3$ is the persistence time in stage 3.

In stage 2, the system is in transition from state $v$ to state $\mu$. Assuming that the transition process is linear, the slope of the process is defined as the rate of change. Based on two points of the process, $A(t_a, x_a)$ and $B(t_b, x_b)$, the parameter $h$ can be expressed as:

$$h = \frac{x_b - x_a}{t_b - t_a}. \tag{5}$$

The parameters $\alpha, \beta$ are defined to describe the two points' locations:

$$\begin{cases} x_\alpha = \alpha(\mu - v) + v \\ x_\beta = \beta(\mu - v) + v \end{cases}. \tag{6}$$

And the solution(as follows) of the model is to describe the points' locations too.

$$t = \frac{1}{\kappa(\mu - v)} \ln\left( \frac{x_0 - \mu}{x_0 - v} \cdot \frac{x - v}{x - \mu} \right) + t_0, \tag{7}$$

Based on Eqs. (5-7), the parameter $h$ can be expressed as:

$$\begin{aligned} h &= \frac{\alpha(\mu - v) - \beta(\mu - v)}{\frac{1}{\kappa(\mu - v)} \left( \ln\left( \frac{\alpha(\mu - v)}{\alpha(\mu - v) - \mu + v} \cdot \frac{\beta(\mu - v) - \mu + v}{\beta(\mu - v)} \right) \right)} \\ &= \kappa(\mu - v)^2 \frac{\alpha - \beta}{\ln\left( \frac{\alpha}{\beta} \cdot \frac{1 - \beta}{1 - \alpha} \right)} = \kappa \omega^2 \chi \end{aligned} \tag{8}$$

Which the new location parameter is defined as $\chi = \dfrac{\alpha - \beta}{\ln\left( \dfrac{\alpha}{\beta} \cdot \dfrac{1 - \beta}{1 - \alpha} \right)}$, and a change amplitude parameter is defined as

$\omega = \mu - v$. Yan *et al* (2015) discovered that $\chi$ varies only slightly when the values of $\alpha, \beta$ fall within a certain range.

And when $\alpha = 0.2, \beta = 0.8$, $\chi \, (= 0.2164)$ is constant.

According to Eq. (3), parameter $h$ can be estimated with the least-squares method:

$$h = \sum_{i=n_1+1}^{n_1+n_2} \bar{t}_i \cdot \bar{x}_i \bigg/ \sum_{i=n_1+1}^{n_1+n_2} \bar{t}_i^{\,2} \,, \tag{9}$$

where $n_2$ is the persistence time in stage 2. Then parameter $\kappa$ can be expressed as:

$$\kappa = \frac{h}{(\mu - \nu)^2 \chi}. \tag{10}$$

Based on Eqs. (4) and (9), parameters $\mu, \nu, h$ can be estimated optimally by changing $n_1, n_2, n_3$, and parameter $\kappa$ can be calculated using Eq. (10).

Note that parameter $\kappa$ is a stability parameter, which means that the larger its value, the more unstable the system becomes. The quantitative relationship among the rate of change $h$, the stability parameter $\kappa$, and the change amplitude $\omega$ is shown in Eq. (8). The relationship between the rate of change $h$ and the stability parameter $\kappa$ is linear, but that between the rate of change $h$ and the change amplitude $\omega$ is parabolic. According to a numerical test which applied this method to several ideal models, these relationships were clearly verified. The test showed that the ratio of the rate of change and the stability parameter is constant when the change amplitude is fixed. Moreover, when the stability parameter is fixed, the ratio of the rate of change and the square of the change amplitude is also constant.

By applying this method to the Pacific decadal oscillation index, Yan et al. (2015) verified its transition process. In present paper, this method is applied to analyze the GSST transition process. It is noticed that the sub-sequence was set as 10a, 20a, 30a, and 40a, and the sub-sequence is set as 10a.

The data used in present paper was reconstructed by the National Oceanic and Atmospheric Administration ( http://www.esrl.noaa.gov/psd/data/gridded/data.noaa.ersst.html). The time span of the monthly data is from January 1854 to November 2012, and the spatial resolution is 2×2°. These data have been determined to be reliable (Hirahara *et al*. 2014; Liu *et al*. 2015; Ratna 2015). During the calculation, the time series of each grid was processed for anomalies by month.

**3 Spatial positions of abrupt changes**

**3.1 Abrupt changes at different start/end moments**

The detection method is applied to identify the temperature sequence in each grid. In order to confirm the climate change, all start moments and end moments were counted. As shown in Fig.1, the frequencies of start moments and end moments are displayed from 1854 to 2010. In order to confirm the number of abrupt changes, a threshold of the frequencies was set by an ideal numerical experiment. In the experiment, each original SST series is replaced by a new time series which is rebuilt by shuffle algorithm(it is a algorithm to disturb a time series randomly). The new random series has the same mean and

variance with the original one. By detecting its abrupt changes with the novel method, the frequencies of start moments and end moments are counted as figure 1, and all frequencies are less than 1%. Then, when the frequencies of start moments in 1878, 1942, 1976 and two periods(1890-1920, 1990-2010) are larger than 1%(Fig.1a), the abrupt changes are considered as the real abrupt changes. While according to end moments(Fig.1b), the frequencies are in 1886, 1950 and 1982 and two periods(1900-1930, 1990-2010). By comparing the two frequencies, a certain corresponding relationship shows that the climate changes starting in periods of Fig.1a and ending in periods of Fig.1b as marked by the red arrows and blue boxes respectively. For the climate change starting in 1878 and ending in 1886, the frequency of former is larger than later. The reason maybe that some grids started to change in 1878, while some of this grids did not end in 1886. The same situation occurs in other periods.

In order to verify the corresponding relationship between the frequencies, the spatial distribution of abrupt change classified based on the start/end moments is shown in Fig.2.

(1) As shown in Fig. a, the abrupt change starting in 1878 occurred mainly in the northern Indian Ocean, part of the central North Pacific and South Pacific, and the equatorial Atlantic region. These regions coincided with those of abrupt change ending in 1886 (Fig. b), which indicates that they belonged to the same ACP.

(2) As shown in Fig. c, the spatial distribution of abrupt change starting in 1942 coincided with that ending in 1950 (Fig. d), mainly covering the coastal regions in the northern and western Indian Ocean and part of the equatorial Atlantic regions. This indicates that they are part of the same ACP.

(3) As shown in Fig. e, the abrupt change starting in 1976 mainly occurred in the middle-eastern equatorial Pacific Ocean and small regions of the South Pacific near the South Pole. These regions completely coincided with those of the abrupt change ending in 1982 (Fig. f), indicating that they were part of the same ACP.

(4) As shown in Fig. g, the abrupt change starting in 1890-1920 may be divided into three principle periods: that in 1890-1898 occurred mainly in the Indian and South Pacific Oceans; that in 1900-1993 occurred mainly in the North Pacific and Atlantic; and the most one in 1908-1909 occurred mainly in the eastern and western regions of the equatorial Pacific. This coincides with the spatial distributions (Fig. h) of abrupt change ending in 1902-1903, 1896-1898 and 1908-1910, and proves that they are part of the same ACP.

(5) As shown in Fig. i, the abrupt change which occurred in 1990-2010 may be divided into three periods: that in 1995-1997 occurred mainly in the western region of the South Pacific; that in 1998-1999 also occurred mainly in the western South Pacific and some regions of the North Pacific; and the most recent one occurred in 2005 and 2007, mainly in the Arctic region. In addition, the abrupt changes starting in 1995-1997 ended in 1997-1999; and that starting in 1998-1999 and 2005-2007 in the Arctic region ended in 2006-2008 (Fig. j). These indicate that they are part of the same abrupt change.

The above analysis verifies that the abrupt change of a certain grid point undergoes three steps: start, sustaining and end. The spatial distribution situation of the entire ACP coincides to a large degree with the distribution situation of the abrupt change "point" proposed by Xiao(2007). This indicates that abrupt changes detected by the traditional "abrupt change point" are contained within the "ACP".

It's obvious that the abrupt changes starting in 1878 and 1942 mainly occurred in Indian ocean. And the Indian Ocean Dipole(IOD) of the two years got strong negative phase and strong positive phase respectively(Suryachandra et al, 2002), which could be the trigger of the abrupt changes. The abrupt change starting in 1976 is known as a climate shift, and most climate elements were detected to experience abrupt change. The abrupt changes of 1890-1920 occurred mainly in Pacific ocean, and it is associated with Interdecadal Pacific Oscillation(IPO) index which transits from negative phase to positive phase during this period. The abrupt changes starting at the end of 2000s mainly occurred in high latitudes, which leads to a significant increase of temperature in the polar.

[revised manuscript text omitted]
. In present paper, by applying a novel transition process detection method, the SST system is verified to be bistable and several abrupt changes are detected. The results also provides an understanding of the transition period, in which the system has already left its original state, but has still not reached its new state

(1) The transition process is a significant period of an abrupt change event. According to the start and end moment, the abrupt change event can be divided into three stages. Based on GSST series, several abrupt changes have been identified in the past 100 years. And the spatial distribution reveals that abrupt change amplitudes in low-latitude regions are small, but those in regions of middle and high latitude are large.

(2) In view of these characteristics of the abrupt change process, the spatial distribution of abrupt change duration is analyzed. It is discovered that abrupt changes in different periods have different durations. The durations of abrupt changes before 1942 were generally long (about 5–10 years); that in 1976 was about 5–8 years; and those after this time were short (about 5 years). It's indicateds that the durations being shortened of abrupt changeis consistnt with global warming.

(3) The spatial distribution of abrupt change amplitudes indicates that the temperature decreased throughout the world mostly before 1942, but mostly increased after that. The statistical results for abrupt change amplitude verifies that the SST is bistable and that the system state variables have shifted many times from one stable state to another.

(4) By analyzing the parameters estimated from the SST system, a quantitative relationship is demonstrated that the relationship between the rate of abrupt change and stability is linear, and the relationship between the rate of abrupt change and change amplitude is quadratic. This observation reveals the cause of the high rate and large amplitude of abrupt climate change.

In present paper,  the transition process of abrupt climate change are detected, basing on reanalysis data from NOAA, and a quentitative relationship is exposed. With the quantitative relationship, a further research about the prediction of the abrupt climate change is deserved to be done.

[revised manuscript text omitted]

**Figure 2: Spatial distribution classified based on frequency of start and end moments of abrupt change. Figures. a and b are the climate changes starting in 1878 and ending in 1886. Figures. c and d are the climate changes starting in 1942 and ending in 1950. Figures. e and f are the climate changes starting in 1976 and ending in 1982. Figures. g and h are the climate changes starting in 1890-1920 and ending in 1990-2010. Figures. i and j are the climate changes starting in 1990-2010 and ending in 1990-2010.**

[Figure]

a: ACP start in 1878
b,c,d: ACP start in 1890-1920
e: ACP start in 1942
f: ACP start in 1976
g,h: ACP start in 1990-2010

**Figure 3: Time sequence average of abrupt changes occurring under the background of the same abrupt change: Figure. a is the abrupt change starting in 1878; Figures. b-d is the abrupt change occurring in 1890-1920, including the following; Figure. e is the abrupt change starting in 1942; Figure. f is the abrupt change starting in 1976; and Figures. g-h is the abrupt change starting in 1990-2010.**

[Figure]

**Figure 4: Spatial distribution of abrupt change duration in different periods: Figure. a is the abrupt change in 1878; Figure. b is the abrupt change in 1896-1898; Figure. c is the abrupt change in 1900-1903; Figure. d is the abrupt change in in 1908-1909; Figure. e is the abrupt change in 1942; Figure. f is the abrupt change in 1976; Figure. g is the abrupt change in 1989-1999; and Figure. h is the abrupt change in 2005-2006.**

[Figure]

**Figure 5: Amplitude distributions of abrupt changes in different periods: Figure. a is the abrupt change in 1878; Figure. b is the abrupt change in 1896-1898; Figure. c is the abrupt change in 1900-1903; Figure. d is the abrupt change in in 1908-1909; Figure. e is the abrupt change in 1942; Figure. f is the abrupt change in 1976; Figure. g is the abrupt change in 1989-1999; and Figure. h is the abrupt change in 2005-2006.**

[Figure]

**Figure 6: Bistability state characteristics study of SST system: Figures. a and b are the system state variables before and after abrupt change, respectively; and Figure. c is the variation quantity in the ACP of the system, namely abrupt change amplitude**

[Figure]

**Figure 7:** Relationships between abrupt change rate $h$ and instability parameter $\kappa$ in different periods and different color represent different degree by abrupt change amplitude parameter. Figures. a-c show the abrupt changes in 1878, 1942 and 1976, respectively; and Figures. d-e show the abrupt changes in 1890-1920 and 1990-2010.

[Figure]

**Figure 8:** Relationship between abrupt change rate $h$ and abrupt change amplitude parameter $\omega$, and different color represent different degree by instability amplitude. Figures. a-c show the abrupt changes in 1878, 1942 and 1976. Figures. d-e are the abrupt changes in 1890-1920, 1990-2010.

---

## Author Response (AR1)

Dear editors,

All comments have been answered in the public discussion, and author's changes in manuscript

are as follows, more details are showing in the marked-up manuscript.

1. The word "and" between authors' name is removed.

2. The word "section"(page 3, line 5) is replaced by "stage" as suggested by Referee #1.

3. In page 4, line 27, one sentence is added to explain the first question of Referee #2 which is about the threshold(1%), and a numerical experiment is added to test the threshold as shown in "author's response to referees 2".

4. According to Referee #2 suggestion, some explanations in terms of climate systems are added in section 3.1(page 6, line 1)

5. Two references are added in the manuscript(page 11, line 13; page 11, line28)

6. Figure1 is replaced by an improved edition(page 12, line 14)

7. Some other changes are showing in the marked-up manuscript (Supplement) clearly.

**Reply to Referee #1**

Dear Referee,
We appreciate your interest to our article and your comments. We reply on the comments as follows:

(1) **Referee.** In figure 4, the abrupt changes occurred in different locations of the oceans at the same time. Please discuss possible mechanism about them. Are there some teleconnections between them or just a casual one?

   **Authors.** Actually, it is a question about the standard of the abrupt change. In present manuscript, the method is based on the percentile threshold method, and this is a relatively standard. Thus, when an abrupt change in one grid is detected, it means that the difference between the two states before and after the abrupt change of this moment has the biggest amplitude by comparing with the other moment. Taking fig. A3 for example, we know that both the two series have two abrupt changes in moment t1 and t2 respectively. However, when we detect the abrupt change by setting a standard, only one abrupt change can be considered as a real abrupt change. The amplitude of the abrupt change should be bigger than any others. Then, it is in moment t1 for grid 1 and it is in moment t2 for grid 2. Therefore, when the abrupt changes of different grids are detected, the abrupt change moments of two adjacent points could be different. It is inferred that the longer the time sequence, the more disperse the abrupt change space will be.

   There could be some teleconnections between the abrupt changes occurring in different locations at the same time, because they are likely to be driven by the same force. It's complex and difficult to explain the mechanism. Before that, it has to dig out how the different standard effect on the detection of abrupt change. When all the real abrupt changes are obtained, it is easy to know how the abrupt change transforming from one place to another in dynamics. However, in present manuscript, we just did some basic research about the transition process, proposing a special quantitative relationship. The relationship will be used to study the physical mechanism of abrupt change by climate model.

(2) **Referee.** Since the parameter χ is a function of α and β, χ becomes a constant if set α=0.2, β=0.8. Why?

   **Authors.** This issue was discussed in the previous paper(Yan et al, 2015). α and β represent two points($A(X_\alpha,t_\alpha)$, $B(X_\beta,t_\beta)$) of the transition process, respectively (as shown in fig. A1), then α, β ∈ (0,1). According to the function as mentioned in line 15, page 3, the location parameter χ can be expressed by α and β. As shown in fig. A2, it is found that parameter χ is almost constant when the values of α and β are within a certain range. Therefore, we think that that parameter χ is approximately constant.

(3)**Referee.** The word "and" should be removed in the author list.

   **Authors.** We are going to correct this mistake.

(4)**Referee.** Formula (3): The word "section1, section2, section3" may be replaced by "stage 1, stage2, stage3" or "domain1, domain2, doman3"?

   **Authors.** The words "section1, section2, section3" will be replaced by "stage 1, stage2,

stage3".

**Appendix:**

**Figures**

[Figure]

Fig. A1. The transition process of abrupt change with two points. (according to reference [1])

[Figure]

Fig. A2. The relationship parameter $\chi$ and parameters $\alpha$, $\beta$, where the x axis is parameter $\alpha$, the y axis is parameter $\beta$ and the contour is parameter $\chi$. (according to reference [1])

[Figure]

Fig. A3. To detect the abrupt change of different grids.

**Reference**

[1] Yan P. C., Feng G. .L, Hou W., 2015: A novel method for analyzing the process of abrupt climate change. Nonlin. Processes Geophys. 22, 249–258, doi,10.5194/npg-22-249-2015

**Reply to Referee #2**

Dear Referee,

We appreciate your interest to our article. As your noticed, the method was published in this same journal, and this manuscript is a following work. In the previous paper, a novel method was proposed and applied to the Pacific Decadal Oscillation index. It's true that the index states "a system that are continually changing its behaviour between positive and negative states". By applying the detection method, several abrupt changes were verified. In reference [1, appendix], this method was used to study the transition process of 500 hPa temperature field. That result exposed a new statistical characteristics of the abrupt climate change.

And for the comments, we reply on them as follows:

1. Referee. PDO index synthesize the collective behaviour of an extended area of the ocean.But having a great number of series varying we can randomly detect a possible abrupt climate change that is not real in a single point of the ocean. To avoid this possibility authors put a threshold of 1% of the points performing this abrupt changes. Nevertheless the value of SST of one grid square is not independent of the value of the SST of neighbouring grid squares. Therefore I suggest to made an interdependence test to calculate this threshold. This can be done by replacing SST series with a Gaussian noise series generated from a normal population whose mean and variance are identical to that of the series over the whole studied period, examining abrupt changes in these series and repeating the process a number of times equal to the number of grid points. In this way the number of abrupt changes can diminish and the results could be more representative of real abrupt changes.

Authors. In figure 1(appendix), a brief sketch map shows how to verify the abrupt change with the percentile threshold method. When the blue box moves to moment i, a start moment(red point) of the abrupt change is verified by fitting the sub-sequence with the piece function(as introduced in the manuscript). Then, by moving the blue box to moment i+1, the same point is identified as the start moment. When the blue box moves on the entire time sequence, a series of start moments would be detected. It's noticed that the amplitudes are obtained when the start moments are identified. Through the percentile threshold method, as shown in right figure, the amplitudes with high value(more than the confidence level) is believed as the real abrupt change. Then the start moment of the abrupt change of this series can be obtained. In figure 1(manuscript), the frequency of all grids' start moments and end moments were shown. When the standard is set as 1%, it means that abrupt change occurs in more than 1% of all global grids at that moment. Apparently, when the standard is set as 2% or more, abrupt changes of some moments will not be displayed for studying in this manuscript, but it would not be diminished. In this manuscript, we set the standard as 1% to study several abrupt changes and demonstrate the quantitative relationship of the transition process.

A numerical experiment was taken to test the method as shown in figure 2(appendix). Two ideal time series and their frequencies of the amplitude are shown. All amplitude's value are less

than the confidence level(2.05), which indicates that there are no abrupt changes for the time series. For each gird time series, the number of abrupt changes is supplemented(supplementary documents ), and they are different from each other.

2. Referee. Once performed the point 1 it is possible that the number of abrupt changes diminishes. In any case I suggest to the authors to made the effort to give an explanation of the patterns in terms of climate systems. Thus for example in figure 1 the changes detected in 1976 and 1982 are related to ENSO area. The change of 1976 is a well documented phenomenon called climate shift. In 1982 one of the most important ENSO episodes took place.

Authors. Actually, as discussed in question 1, the number of abrupt changes won't be diminished.

3. Referee. As a minor question I also suggest to carefully read the paper because there are some important mistakes.

Authors. The authors thank the anonymous Referee #2, and the mistakes will be corrected.

**Appendix:**

[Figure]

Fig.1. For each time series, a brief sketch map to tell how to verify the abrupt change with the percentile threshold method.

[Figure]

Fig.2. Two ideal time series and their frequencies of amplitude.

**Reference**

[1]Yan Pengcheng, Feng Guolin, Hou Wei, et al. 2014. Statistical characteristics on decadal abrupt change process of time sequence in 500 hPa temperature field [J]. Chinese Journal of Atmospheric Sciences (in Chinese), 38 (5): 861−873.